# Interaction of SARS-CoV-2 Nucleocapsid Protein and Human RNA Helicases DDX1 and DDX3X Modulates Their Activities on Double-Stranded RNA

**DOI:** 10.3390/ijms24065784

**Published:** 2023-03-17

**Authors:** Camilla Lodola, Massimiliano Secchi, Virginia Sinigiani, Antonella De Palma, Rossana Rossi, Davide Perico, Pier Luigi Mauri, Giovanni Maga

**Affiliations:** 1Institute of Molecular Genetics IGM CNR “Luigi Luca Cavalli-Sforza”, Via Abbiategrasso 207, 27100 Pavia, PV, Italy; 2Institute of Biomedical Technologies ITB-CNR, Via Fratelli Cervi 93, 20054 Segrate, MI, Italy

**Keywords:** SARS-CoV-2, dead-box RNA helicase, RNA binding, nucleocapsid, DDX3X, DDX1

## Abstract

The nucleocapsid protein Np of SARS-CoV-2 is involved in the replication, transcription, and packaging of the viral genome, but it also plays a role in the modulation of the host cell innate immunity and inflammation response. Ectopic expression of Np alone was able to induce significant changes in the proteome of human cells. The cellular RNA helicase DDX1 was among the proteins whose levels were increased by Np expression. DDX1 and its related helicase DDX3X were found to physically interact with Np and to increase 2- to 4-fold its affinity for double-stranded RNA in a helicase-independent manner. Conversely, Np inhibited the RNA helicase activity of both proteins. These functional interactions among Np and DDX1 and DDX3X highlight novel possible roles played by these host RNA helicases in the viral life cycle.

## 1. Introduction

The coronavirus SARS-CoV-2 is the causative agent of the current pandemic of COVID-19. Following a zoonotic spillover from an as yet unknown animal reservoir in mainland China in 2019, SARS-CoV-2 rapidly spread all over the world. The accelerated development of vaccines, monoclonal antibodies and antivirals has provided an effective means to counteract the most severe consequences of the infection. However, the efficacy of the currently available vaccines and antibodies is being challenged by the rapid emergence of SARS-CoV-2 variants, carrying immune escape mutations on the Spike glycoprotein, the major viral antigen. Thus, it is essential to develop alternative approaches which can counteract SARS-CoV-2 infection. To this aim, developing a deeper understanding of the interplay among viral and host proteins is of great importance. The positive sense single-stranded (ss) RNA genome of SARS-CoV-2 comprises 14 open reading frames (ORFs), encoding four structural and 16 non-structural proteins. The ninth ORF encodes for the nucleocapsid protein (Np). Np is highly conserved among the genus coronavirus and represents one of the most abundant structural proteins expressed in SARS-CoV-2 infected cells. Np consists of 419 aminoacids (aa) and it is organized into two structured modules, the N-terminal domain (NTD) and the C-terminal domain (CTD), which are connected by the intrinsically disordered region (IDR) Ser-/Arg-rich flexible linker (LKR). Two additional IDRs, the N-arm (aa 1–43) and the C-tail (aa 365–419), are present at the extreme N- and C-terminal parts of the protein, respectively [1]. While the NTD and CTD isolated domains have been crystalized, the presence of the IDRs makes the resolution of the structure of the full-length Np a challenging task. A combination of structural studies, in silico modeling and small-angle X-ray scattering revealed that the NTD, CTD and LKR domains are involved in RNA binding [2]. Molecular modeling suggested that Np can bind both ss and dsRNA. Electron microscopy analysis, combined with dynamics simulations, suggested that in its unbound state Np exists as a dimer in solution with the NTD and CTD domains in an extended conformation. Conversely, the binding of RNA induces a compact conformation and triggers the cooperative formation of higher-order Np–RNA oligomeric complexes such as tetramers, hexamers or octamers [3,4,5,6].

The ability of Np to form protein-RNA filaments is essential for the packaging of the viral genomic (g) RNA into the new virions [1]. However, coronavirus Np has been also shown to participate in the genome transcription process, facilitating the synthesis of subgenomic (sg) viral mRNAs by binding to regulatory gRNA regions [7,8]. In addition, Np can modulate the host cell metabolism, suppressing the host innate immune response [9,10]. Np interacts with the host receptor RIG-I, preventing its binding to the E3 ligase TRIM25 [11], and with the STAT1/2 transcriptional activators, inhibiting their phosphorylation by the JAK1/TYK2 kinases and subsequent nuclear translocation [12]. Thanks to these interactions, Np acts as a potent antagonist of the IFN response. Np can also inhibit the formation of viral-induced stress granules, a host defense mechanism, by sequestering the proteins G3BP1/2 [13,14].

A central role in the host IFN response is played by the cellular DEAD-box RNA helicases DDX3X and DDX1 [15]. DDX3X binds to viral dsRNA and recruits the RIG-I-stimulating factor IPS-1, promoting IFN-b production [16]. In addition, it is ubiquitinated by the E3 ubiquitin ligase TRIM25, resulting in further stimulation of the IFN-b response [17]. DDX1 also binds dsRNA as part of a viral RNA sensor complex, together with the RNA helicases DDX21 and DHX36 and the adaptor protein TRIF, which triggers the IFN-b response [18]. Besides their roles in innate immunity, DDX3X and DDX1 are involved in many aspects of RNA metabolism, from mRNA synthesis, maturation and nuclear export to miRNA biogenesis and protein translation [19]. Several RNA viruses encode viral proteins, which physically interact with these RNA helicases. Such a hijacking strategy has a dual role: on the one hand, to prevent DDX3X and/or DDX1 from activating the innate immune response, and, on the other hand, to exploit these RNA helicases for promoting viral genome replication and transcription, as well as viral protein expression. For these reasons, DDX3X and DDX1 are considered promising targets for antiviral therapy [20,21].

The Np of different coronaviruses has been shown to physically interact with both DDX3X and DDX1 [22]. Indeed, through immunoprecipitation and colocalization studies in infected cells, SARS-CoV-2 Np has been shown to interact with both DDX3X and DDX1 [23].

However, the available data did not allow us to determine whether the interaction of SARS-CoV-2 Np with DDX3X and DDX1 was direct or instead mediated by other factors such as RNA or proteins. In addition, the functional consequences of these interactions with the biochemical properties of either the RNA helicases or Np are currently not yet clarified.

Here, we show that ectopic expression of SARS-CoV-2 Np induces significant changes in the proteome of the human A549 lung adenocarcinoma cell line. DDX1 was one of the proteins that was most enriched upon Np expression. About 60% of the differentially expressed proteins found in our analysis have been also previously identified as part of the Np interactome of SARS-CoV-2 Np in the Calu-3 lung cell line. Interestingly, several of the Np interactors present in our dataset are known interactors of either DDX1 or DDX3X as well. Most notably, DDX1 is listed among the interactors of both Np and DDX3 and it was enriched in our analysis upon Np ectopic expression in A549 lung carcinoma cells.

Considering that DDX3X and DDX1 interact with hundreds of other cellular proteins, the Np-DDX1-DDX3X reciprocal interaction might represent an important node in the Np-mediated network of interactions with the host cell proteome. Thus, we investigated the possible functional consequences of such interactions on their common substrate dsRNA. The expression and purification of recombinant SARS-CoV-2 Np and human DDX3X and DDX1 allowed us to confirm their direct physical interaction. Biochemical characterization of these proteins, either alone or in combination, showed that they reciprocally modulated their interaction with dsRNA.

These findings are discussed in the context of the possible roles played by DDX1 and DDX3X in the viral life cycle.

## 2. Results

### 2.1. Expression of the SARS-CoV-2 Np Protein Induces Significant Alterations in the Proteome of A549 Human Cells

To investigate the expression variations in A549 cells transfected with SARS-CoV-2 Np or empty vector, we performed a differential proteome analysis with a label-free shotgun approach based on nano liquid chromatography coupled to high-resolution tandem mass spectrometry (nLC-hrMS/MS).

In total, 4306 master proteins were identified with at least one unique peptide and mapped to 4170 unique gene symbols, acquiring three biological replicates in triplicate runs for the two examined conditions. The application of linear discriminant analysis (LDA [24]) to the 18 protein lists allowed us to extract statistically significant proteins (F ratio ≥ 4.5 and *p* < 0.05) which could then be processed by hierarchical clustering, highlighting a clear separation in the two main branches and the different behaviour of A549 cells expressing when either an empty vector or SARS-CoV-2 Np, as reported in Figure 1.

Then, all proteins identified in single runs were aligned to create a unique list for each experimental group, normalizing and averaging [25] the peptide spectrum match values (aPSMs) attributed to the proteins, which represent the number of mass spectra assigned to each and also indirectly demonstrate their abundance in the samples (see Appendix A).

Using the homemade algorithm MAProMa and applying its two algorithms, DAve (Differential Average) and DCI (Differential Confidence Index), to the aPSMs of each single protein between the two compared terms, it was possible to perform a differential analysis and to quantitatively examine the proteomic changes due to the expression of SARS-CoV-2 Np. In Figure 2, the bar charts show the distribution of up-regulated proteins with a variation greater than a fold change of 1.5 in the two examined conditions according to their DAve values. Proteins with a DAve (ratio of protein expression) ≥ |0.35| and a DCI (confidence of differential expression) ≥ |5| pass the filters and could be considered differentially expressed in the considered comparison. Moreover, to emphasize highly expressed proteins extracted in each of the two compared groups, Voronoi graphs (Appendix A) in which the polygons and their different sizes represent, respectively the involved functional categories and their degree of enrichment have been constructed.

Combining LDA and MAProMa results, 198 differentially expressed proteins (DEPs) were extracted between A549 cells transfected with the empty vector and SARS-CoV-2 Np (see Appendix A).

In particular, 93 proteins were up-regulated in A549 cells expressing SARS-CoV-2 N protein and 105 were up-regulated in A549 cells expressing an empty vector. Notably, proteins that were up-regulated in the first condition of comparison are to be considered down-regulated in the second one and vice versa.

With the aim of achieving better visualization of the pathways and the biological relevant interactions, the STRING database [26] was consulted in order to build a network of both known and predicted protein–protein interactions. Those proteins that were differentially expressed, were represented as nodes, being grouped into sub-networks based on their molecular function and connected by grey edges indicating protein–protein interactions. Figure 3 shows the PPI network obtained by the comparison of the two A549 cell types.

The comparison of the 198 differentially expressed proteins (DEPs) between A549 cells transfected with empty vectors and SARS-CoV-2 Np-expressing cells, with the published SARS-CoV-2 N-interacting proteins in the lung cell line Calu-3 [27], revealed an overlap of 123/198 (62%) proteins present in both datasets (Appendix A) and that they share 37/198 hits (18.7%) with known Np interactors from the BioGRID database [28]. Notably, DDX1 is significantly enriched in A549 cells expressing Np (Figure 2) and its presence is also reported among the DDX3X interactors in the STRING database (https://string-db.org/, last accessed on 3 March 2023).

Considering that DDX1 and DDX3X interact with hundreds of cellular proteins, this analysis suggests that the Np-DDX1-DDX3X reciprocal interaction might constitute an important node in the complex network of interactions of Np with the host cell proteome.

### 2.2. Recombinant Full-Length SARS-CoV-2 Np Protein Cooperatively Binds Partially to dsRNA with High Affinity

In order to investigate a possible functional interaction between Np and the cellular RNA helicases DDX1 and DDX3X, full-length SARS-CoV-2 Np was expressed in *E. coli* and purified (Appendix A). The recombinant protein was able to bind ssRNA, as revealed by EMSA experiments (Figure 4A), indicating that it was fully functional. The optimal substrate of DDX1 and DDX3X RNA helicases is dsRNA with a short 5′-protruding ss end. SARS-CoV-2 Np has been shown to bind structured partially dsRNA regions such as stem-loops with high affinity [6]. Thus, the binding of recombinant Np to dsRNA was assessed by EMSA. A fluorescently labelled 18/38 mer partially dsRNA oligonucleotide with a 5′-protruding end was used, which is the preferred substrate for the DDX1 and DDX3X RNA helicases. Np was able to shift the partially dsRNA substrate (Figure 4B, lanes 5–10; Figure 4C, lanes 2–7), generating signals with different electrophoretic mobilities (Figure 4B,C, indicated by asterisks), as is consistent with the formation of protein–RNA complexes of increasing molecular weight. Quantification of the signals allowed us to reveal a cooperative binding mode, consistent with previous results [3], and an apparent binding affinity of 0.17 µM (Figure 4D). Next, the interaction of Np with the partially dsRNA substrate was tested in the presence of increasing NaCl concentrations in a physiological range of 25–200 mM (Figure 4E, lanes 7–10). No significant reduction of Np–RNA interactions was observed, even at the highest salt concentration (Figure 4F). Thus, recombinant Np was able to bind cooperatively partially to dsRNA with high affinity in a salt-resistant manner.

### 2.3. SARS-CoV-2 Np Protein Physically Interacts with Human DDX3X and DDX1 RNA Helicases in the Absence of RNA

Both DDX1 and DDX3X have been suggested to interact with SARS-CoV-2 Np in cells through co-IP and IF experiments [23]. However, such approaches did not assess whether the interaction was direct or mediated by additional factors, either by proteins or RNA. Thanks to the availability of all three recombinant proteins, here their direct interaction was tested in pull-down experiments in the absence of other factors. In a first set of experiments, HALO-tagged recombinant full-length human DDX3X, bound to HALO affinity beads, was incubated with recombinant Np. Subsequent elution of the pulled-down material by TEV protease digestion of the HALO tag released both proteins, as revealed by Western blot with specific antibodies (Figure 5A,B, lane 5, 6). Performing a mock reaction with Np alone incubated with the Halo tag captured on the beads as a negative control did not result in its retention (Figure 5B, lane 5). In a second set of experiments, MBP-tagged Np bound to MBP affinity beads was incubated, together with purified recombinant DDX1. Subsequent elution of the pulled-down material by TEV protease digestion of the MBP tag released both proteins, as revealed by Western blot with specific antibodies (Figure 5C,D, lane 5). A mock reaction with DDX1 alone, incubated with the MPB tag captured on the beads as a negative control, did not result in its retention (Figure 5C, lane 4).

Collectively, these results indicated that SARS-CoV-2 Np physically interacts with both DDX1 and DDX3X in the absence of RNA or additional factors.

### 2.4. Human DDX1 and DDX3X Increase the Affinity of SARS-CoV-2 Np Protein for Partially dsRNA in a Helicase Independent Manner

In preliminary experiments, the combination of Np with DDX3X or DDX1 apparently increased the amounts of Np–RNA complexes (Appendix A). In order to more precisely investigate the mechanism of Np–RNA complex stimulation by the two RNA helicases, Np titrations were performed in the presence of partially dsRNA and in the absence or presence of a fixed amount of either DDX3X or DDX1 and analyzed in EMSA assays. Increasing amounts of Np alone resulted in a proportional increase in Np–RNA complexes (Figure 6A,B, lanes 2–5 and Appendix A). The addition of a fixed amount of DDX3X (Figure 6A, lanes 6–9 and Appendix A) or DDX1 (Figure 6B, lanes 6–9 and Appendix A) caused increases in the high molecular weight complexes. Quantification of the shifted products in two independent experiments allowed us to perform the analysis of the binding kinetics. The best fitting was obtained with a cooperative binding mode equation. As shown in Figure 6C,D, Np alone showed a Hill coefficient of 1.2–1.4 and an apparent dissociation constant for dsRNA K_D_ of 0.17–0.22 µM, which are consistent with the results shown in Figure 4D. The addition of DDX3X resulted in a K_D_ of 0.08 µM (Figure 6C), while the addition of DDX1 resulted in a K_D_ of 0.04 µM (Figure 6D). Thus, both RNA helicases increased 2- to 4-fold the apparent affinity of Np for the dsRNA.

Next, Np was titrated in the presence of a DDX3X catalytically dead mutant, where the two aspartic acids of the Walker B D-E-A-D motif had been mutated to alanines (mutant DDX3X_DADA_). These mutants are devoid of ATPase and helicase activities [29]. The DDX3X_DADA_ mutant was still able to stimulate the RNA binding activity of Np (Figure 6E, compare lanes 2–6 with lanes 7–10), increasing 3- fold its apparent affinity, as shown by the decreasing K_D_ value from 0.3 µM to 0.1 µM (Figure 6F). Overall, these data indicated that DDX3X and DDX1 were able to stimulate the binding affinity of Np to dsRNA and that the helicase activity was not necessary for such stimulation.

### 2.5. SARS-CoV-2 Np Protein Inhibits the RNA Helicase Activity of Human DDX3X and DDX1

DDX1 and DDX3X are endowed with the ability to unwind dsRNA in an ATP-dependent manner. Thus, the effects of Np on the helicase activities of both enzymes were investigated in a helicase assay in the presence of ATP and the fluorescently labelled 18/38 mer partially dsRNA substrate and the products analyzed by native PAGE. As expected, Np did not show any RNA helicase activity (Figure 7A,B, lanes 2–4). Conversely, displacement of the labelled 18 mer strand was observed with both DDX3X and DDX1 (Figure 7A,B, lane 5). When DDX3X or DDX1 were incubated in the presence of Np, however, inhibition of their helicase activity was observed, as indicated by the decrease in the displaced products and the increase in the unwound substrate (Figure 7A,B, compare lane 5 with lanes 6–8).

These results indicated that association of Np with DDX3X or DDX1 represses their RNA unwinding activity.

Thus, Np, DDX3X and DDX1 reciprocally influence their respective interaction with RNA, but in different ways.

## 3. Discussion

Besides driving the packaging of the viral genome, SARS-CoV-2 Np plays additional important roles in several aspects of the viral life cycle. Several studies showed that Np undergoes liquid–liquid phase separation (LLPS) to form protein–RNA compartments in the cytosol. These have been shown to enhance recruitment of viral and host proteins in ribonucleoprotein complexes during the replication and transcription of the viral gRNA [30,31]. Transcription of the viral genome generates sgmRNAs through a template switching mechanism from internal transcription regulatory sequences (body TRS), located in front of each gene, to the leader TRS located at the 5′-end of the genome [8,32]. The extensive stem-loop structures, present in the gRNA among the body TRS, have been demonstrated to be engaged in long-range RNA–RNA interactions critical for template switching [33]. SARS-CoV-2 Np was shown to bind TRS, probably facilitating long distance interactions between differently structured gRNA regions thanks to its ability to form multimers and undergo LLPS [34]. Interestingly, the JHMV-CoV Np was found to interact, upon phosphorylation, with the RNA helicase DDX1. This Np/DDX1 complex was shown to regulate the synthesis of long sgmRNAs by modulating the rate of template switching from the different body TRS encountered by the transcription complex. Interestingly, only the overexpression of the wild type, but not of an inactive mutant of DDX1, promotes the synthesis of long sgmRNAs, suggesting that the enzymatic activity of DDX1 is important for stimulation of viral RNA synthesis and could maybe help to resolve RNA secondary structures [35].

In this work, applying a label-free shotgun proteomic approach, we were able to show that a significant alteration in A549 human cells is induced by SARS-CoV-2 Np expression. A great number of differentially expressed proteins, resulting from the comparison of A549 cells transfected with empty vector and SARS-CoV-2 Np, are already known as SARS-CoV-2 Np interactors. In fact, the PPI network built based on these proteins highlights deregulated the pathways involved in RNA processes (splicing and translation), immune response, cytoskeleton organization and mitochondrial metabolisms, as emerged from previous studies, despite the use of different techniques and cell lines.

Moreover, we provide the first evidence, to the best of our knowledge, that SARS-CoV-2 Np also directly interacts with DDX1 in its unphosphorylated form and that such interaction is not mediated by other components, whether RNA or proteins. The Np–DDX1 complex shows a higher affinity for dsRNA than Np alone while also suppressing the DDX1 RNA helicase activity. In the context of the template-switching mechanism discussed above, it can be hypothesized that DDX1 can be recruited at the viral transcription complex, maybe through its interaction with Nsp14, as reported for IBV-CoV and SARS-CoV [22], helping the RdRP to overcome secondary structures. The advancing transcription complex might then encounter Np which is bound to the structured region around the TRS located in proximity of the 5′-end of the genome. There, interaction between Np and DDX1 will inhibit the RNA helicase activity, leading to the formation of a stable Np–DDX1 complex, which could function as a signal for the termination of viral RNA synthesis. Indeed, in JHMV-CoV, it has been shown that the Np/DDX1 complex has higher affinity for the regions proximal to the 5′-end TRS (with the exception of the 5′-terminal leader TRS). The Np/DDX1 complex could then mediate the template switching from the body TRS to the leader TRS. The ability of Np to undergo multimerization and LLPS in the presence of dsRNA may facilitate the genome cyclization, which has been proposed as exposing the leader TRL for base pairing with the nascent sgmRNA [33]. We have shown that DDX1 interaction does not prevent Np multimerization on dsRNA. Thus, the DDX1–Np complex might provide a stable platform for the pairing of the body TRS with the leader TRS thanks to its high affinity for dsRNA, at the same time preventing the DDX1 RNA helicase activity from interfering with the template switching.

Np is also an important modulator of the host immune response [9,10]. SARS-CoV-2 Np inhibits the dsRNA recognition by the RIG-I receptor, preventing with its ubiquitination the binding of the E3 ligase TRIM25 and inhibiting the formation of antiviral stress granules (SGs) through its interaction with the G3BP1 protein [11,13]. In addition, Np can directly bind to RIG-I, preventing the downstream signaling cascade and leading to IRF3 phosphorylation and nuclear translocation [14]. All these mechanisms repress IFN-b production. RIG-I binding of dsRNA leads to the activation of the IPS-1 (MAV) adaptor protein located on the outer mitochondrial membrane. This in turn mobilizes the kinase complex TBK1/IKKe, phosphorylating the transcription factor IRF3 which leads to IFN-b production [36]. DDX3X has been shown to be an essential component of this pathway. DDX3X interacts with TRIM25 and can stimulate IFN-b production by directly binding with dsRNA and forming a complex with IPS-1. After this, it acts further downstream, associating with the TBK1/IKKe complex, enhancing its activity, and also functioning as a transcriptional coactivator of the *IFNB1* gene promoter [17]. In this context, our results might suggest that Np can sequester DDX3X in a stable complex with dsRNA, preventing its interaction with IPS-1. In addition, by binding with DDX3X, Np might prevent its association with the TBK1/IKKe and its nuclear binding to the *IFNB1* promoter, resulting in a potent inhibition of the IFN-b response, analogously to what has been observed for other viruses.

DDX3X is also essential for the formation of antiviral SGs, large ribonucleoprotein complexes which can sequester viral mRNAs, suppressing their translation [37]. DDX3X also regulates inflammasome activation by the NLRP3 protein. DDX3X interacts with NLP3 to induce inflammasome complex formation, after which DDX3X also stimulates SG formation. Sequestration of DDX3X within SGs competes for its binding to NLRP3, thus reducing inflammasome activation [38,39]. Notably, SARS-CoV-2 Np was shown to directly bind to NLRP3, inducing a strong inflammatory response through inflammasome activation [40]. Thus, by sequestering DDX3X, Np can prevent antiviral SGs formation, at the same time replacing DDX3X in the interaction with NLRP3 in order to maintain inflammasome activation. Interestingly, DDX3X has been proposed to act by promoting the LLPS of ribonucleoprotein complexes [41]. Thus, its sequestration by Np might also stimulate the Np LLPS activity.

Finally, DDX3X inhibition by small molecules or siRNA was shown to suppress SARS-CoV-2 replication, suggesting a proviral role [42,43]. While the mechanistic details of such a role remain so far elusive, an intriguing hypothesis is that DDX3X might be re-directed through interaction with Np to the lipid structures where the viral replication/transcription complex (RTC) is located. A similar redistribution of DDX3X at sites of viral replication has been shown for DDX3X in the context of HCV infection [44].

## 4. Materials and Methods

### 4.1. Protein Extraction and Enzymatic Digestion

The proteomic analysis described in this study was performed on A549 cells transfected with the empty vector pCDNA3.1(+)N-EGFP (Genscript, Piscataway, NJ, USA) or with the same cells albeit those expressing the Np protein (pCDNA3.1(+)N-EGFP-NP). Expression at 72 h after transfection was verified by Western blot (Appendix A). For each condition, about 2 × 10^6^ cells were used and three biological replicates were examined. After determining the protein concentration with the SPN^TM^-Protein assay kit (G-Biosciences, St. Louis, MO, USA), 50 µg of the resulting suspensions were lysed, reduced/alkylated and enzymatically digested with Trypsin and Lys-C using the Easy Pep^TM^ Mini MS Sample Prep Kit (Thermo Scientific, Rockford, IL, USA). Following the kit protocol, in less than 3 h and for each examined condition, peptides were generated, cleaned-up to prepare detergent-free samples and resuspended in 0.1% formic acid (Sigma, St. Louis, MO, USA) for LC-MS/MS analysis.

### 4.2. LC-MS/MS Analysis

Peptide mixtures were analyzed using an Eksigent nanoLC-Ultra^®^ 2D System (Eksigent, part of AB SCIEX, Dublin, CA, USA) configured in trap-elute mode. Briefly, samples (0.8 µg injected) were first loaded onto a trap (200 µm × 500 µm ChromXP C18-CL, 3 µm, 120 Å) and washed with the loading pump, which was running in isocratic mode with 0.1% formic acid in water for 10 min at a flow of 3 µL/min. The automatic switching of autosampler 10-port valve then eluted the trapped mixture onto a nano reversed phase column (75 µm × 15 cm ChromXP C18-CL, 3 µm, 120 Å) through a 162 min gradient of eluent B (eluent A, 0.1% formic acid in water; eluent B, 0.1% formic acid in acetonitrile) at a flow rate of 300 nL/min. In depth, the gradient was: from 5–10% B in 3 min, 10–30% B in 139 min, 30–95% B in 12 min and holding at 95% B for 8 min. The eluted peptides were directly analyzed on an LTQ-OrbitrapXL mass spectrometer (Thermo Fisher Scientific, Waltham, MA, USA) equipped with a nanospray ion source. The spray capillary voltage was set at 1.7 kV and the ion transfer capillary temperature was held at 220 °C. Full MS spectra were recorded over a 400–1600 *m/z* range in positive ion mode, with a resolving power of 60,000 (full width at half-maximum) and a scan rate of 2 spectra/s. This step was followed by five low-resolution MS/MS events that were sequentially generated in a data-dependent manner on the top five ions selected from the full MS spectrum (at 35% collision energy), using dynamic exclusion of 0.5 min for MS/MS analysis. The mass spectrometer scan functions and high-performance liquid chromatography solvent gradients were controlled by the Xcalibur data system version 1.4 (Thermo Fisher Scientific, Waltham, MA, USA).

### 4.3. Data Handling

All data generated were searched using the Sequest HT search engine contained in Proteome Discoverer software, version 2.1 (Thermo Fisher Scientific, Waltham, MA, USA). Experimental MS/MS spectra were compared with the theoretical mass spectra obtained by the in silico digestion of 29 SARS-CoV-2 protein sequences obtained from Uniprot (www.uniprot.org (accessed on 23 March 2021)) and from the *Homo Sapiens* proteome database (74,842 entries). The following criteria were used for the identification of peptide sequences and related proteins: trypsin and Lys-C as enzymes, methionine oxidation, carbamidomethyl at cysteins, three missed cleavages per peptide, with mass tolerances of ±50 ppm for precursor ions and ±0.8 Da for fragment ions. Percolator node was used with a target-decoy strategy to give a final false discovery rate (FDR) at a peptide spectrum match (PSM) level of 0.01 (strict) based on q-values, considering a maximum deltaCN of 0.05 [45]. Only peptides with a minimum peptide length of six amino acids and rank 1 were considered. Protein grouping and a strict parsimony principle were applied. Results were then exported to an Excel file for further processing.

### 4.4. Differential Expression, Linear Discriminant Analysis and PPI Network

The 18 protein lists (A549 cells transfected with an empty vector and with SARS-CoV-2 N protein), obtained from the SEQUEST algorithm, were processed by linear discriminant analysis (LDA) and proteins with the largest F ratio and smallest *p* value (<0.05) were retained. Ward’s method and the Euclidean’s distance metric were applied using the JMP 15.2 software. Then, all protein lists were aligned, normalized with the total signal normalization method and compared by means of the average peptide spectrum matches (aPSM) [46,47], corresponding to the average of all the spectra identified for a protein and, consequently, to its relative abundance in each analyzed condition. In depth, to select differentially expressed proteins, A549 cells transfected with an empty vector and with SARS-CoV-2 N protein were compared, applying a threshold of 0.35 and 5 on the two MAProMa indexes, DAve (Differential Average) and DCI (Differential Confidence Index), respectively [48]. DAve, which evaluates changes in protein expression, was defined as (X − Y)/(X + Y)/0.5, with values ranging from a maximum of +2.00 to a minimum of −2.00, while DCI, which evaluates the confidence of differential expression, was defined as (X + Y) x (X − Y)/2, with values ranging from +∞ to −∞. The X and Y terms represent the aPSMs of a given protein in two compared samples. Visualization of proteome enrichment was obtained using interactive Proteomaps (available at www.proteomaps.net accessed on 22 October 2022) [49] that automatically built Voronoi graphs from proteome data and based on the KEGG pathways gene classification. Each protein is shown by a polygon and functionally related proteins are arranged in common regions whose size is a function of the abundance of the proteins themselves.

Starting from the DEPs, a protein–protein interaction (PPI) network (197 nodes and 3393 edges) was built by STRING database [50]; only experimentally and database-defined PPIs with a score >0.15 were considered. The resulting sub-network was visualized and analyzed by Cytoscape v. 3.9.1 and its plugins [51]. The proteins were grouped into functional modules by the support of STRING enrichment, using default settings. Node colors reflected the differential expression of DEPs in the two examined conditions.

### 4.5. Plasmid Construction and Protein Purification

The SARS-CoV-2 Np DNA sequence was amplified from plasmid pCVM3-Np (SinoBiological, Eschborn, Germany) with specific primers, namely forward (5′-CGCGGATCCAGTGACAATGGACCACAGAACCACAACCAG-3′) and reverse (5′-ACTCGAGCGGCCGCTCTAGATCAAGCC-3′) (Eurofins Genomics, Ebersberg, Germany) and cloned BamHI-NotI in a pET30a(+) empty vector. Recombinant Np protein was expressed overnight in BL21(DE3) *E. coli* cells with 0.2 mM IPTG at 37 °C and purified mainly as described in [52] with few modifications. Briefly, the bacterial cell pellet was resuspended in a low salt buffer (20 mM Tris pH 8, 20 mM KCl, 0.5 mM EDTA, protease inhibitor) and lysed by adding lysozyme 1 mg/mL and using sonication.

After centrifugation at 20,000 rpm for 30 min, the clear supernatant was sequentially incubated with Q-Sepharose fast flow and SP-Sepharose fast flow resins (GE Healthcare, Uppsala, Sweden) for 2 h and with the Np protein eluted from SP resin using 200–1000 mM KCl gradient at pH 8. Then, the 500–1000 mM fractions were pooled and incubated with Ni-NTA resin (Qiagen, Hilden, Germany) for 2 h at 4 °C. Finally, recombinant Np protein was recovered using a 50–250 mM imidazole gradient and the 100 mM imidazole fractions containing the Np purified protein were pooled, dialyzed and stored at −80 °C, with a final concentration of 10% glycerol. The Np DNA sequence was subcloned BamHI-NotI in pET HIS6-MBP-TEV empty vector (Addgene, Watertown, MA, USA) to obtain the plasmid pET HIS6-MBP-TEV-Np necessary for the expression of MBP-Np protein in a pull-down assay. Human recombinant proteins DDX3X and DDX3X_DADA_ (D347A/D350A) mutant used in helicase and EMSA assays were obtained from plasmids pET30a(+)-DDX3X and pET30a(+)-DDX3X_DADA_, respectively. All the proteins were purified from ShuffleT7 *E. coli* soluble fractions, as already described [53]. Recombinant DDX1 was obtained from the pET30a(+)-DDX1 plasmid with the same purification procedure as described for DDX3X [32]. The pH6HTN-His6-Halotag-DDX3X plasmid was used for the expression of the HALO-DDX3X protein in a pull-down experiment. For plasmid construction, the DNA sequence of DDX3X was amplified from pET30a(+)-DDX3X with specific primers, namely forward (5′-GCTAGGATATCAATGAGTCATGTGGCAGTGGAAAA-3′) and reverse (5′-ATATAGCGGCCGCTCAGTTACCCCACCAGTCAACC-3′) (Eurofins Genomics, Ebersberg, Germany) and cloned EcoRV-NotI in a pH6HTN-His6-Halotag empty vector (Promega, Madison, MD, USA).

### 4.6. Western Blotting

Western blot analysis was performed according to standard procedures. Protein samples were separated by 10% SDS-PAGE, blotted onto Protran-BA83 nitrocellulose membranes (Schleicher & Schuell, Keene, NH, USA) and incubated overnight with mouse monoclonal anti-SARS-CoV-2 Np (1:1000) (SinoBiological, Eschborn, Germany), polyclonal rabbit anti-DDX3X A300-474A (1:1000) (Bethyl, Montgomery, TX, USA) and polyclonal rabbit anti-DDX1 (1:1000) (Bethyl, Montgomery, TX, USA), followed by 1 h of incubation with HRP-conjugated secondary antibodies known as goat anti-mouse or goat anti-rabbit (1:5000) (Sigma, St. Louis, MO, USA). Chemiluminescent signals were developed using the ECL reagent (GE Healthcare, Uppsala, Sweden) and Chemidoc imaging system (Bio-Rad, Hercules, CA, USA).

### 4.7. Nucleic Acid Substrates

RNA oligonucleotides were purchased from Biomers.net GmbH (Ulm, Germany). The sequences of the substrates used are:

ssRNA 38 mer

5′-AUGAAGGUUUGAGUUGAGUGGAGAUAGUGGAGGGUAGU-3′

ssRNA 18 mer FAM

3′-UACUUCCAAACUCAACUC-5′ FAM

For dsRNA, oligonuclotides were mixed at 1:1 (M/M) ratio in an annealing buffer (30 mM HEPES–KOH pH 7.4, 100 mM KCl, 2 mM MgCl_2_, 50 mM NH_4_Ac) at a final concentration of 500 nM, heated at 95 °C for 5 min and then slowly cooled down at room temperature.

### 4.8. Electrophoretic Mobility Shift Assay (EMSA)

EMSA assays were performed by incubating 50 nM ssRNA 18 mer FAM or 50 nM dsRNA 18/38 mer FAM with the different concentrations of the purified proteins for 1 h at 4 °C in an EMSA buffer (20 mM Tris–HCl pH 8, 2 mM DTT, 70 mM KCl, 2 mM MgCl_2_, 6U RNasin, 5% Glycerol). When Np was used in the presence of DDX1, DDX3X and DDX3X_DADA_ proteins were preincubated for 10 min at 4 °C before the addition of the dsRNA substrate. Reactions were stopped by adding 6x gel loading buffer (Thermo Fisher Scientific, Waltham, MA, USA). Then, samples were separated by 12% TBE-polyacrylamide (19:1 from SERVA Electrophoresis GmbH, Heidelberg, Germany) gel at 40 V for about 3 h in a 1x TBE buffer at 4 °C in a mini-PROTEAN electrophoresis system (Bio-Rad, CA, USA). Substrates and products were quantified by laser scanning densitometry at Typhoon-TRIO (GE Healthcare, Uppsala, Sweden).

### 4.9. Pull-Down Assay

Pull-down assays were performed to verify the interaction between HALO-DDX3X with Np protein and MBP-Np with DDX1 using a similar procedure. HALO-DDX3X and MBP-Np proteins were expressed in BL21(DE3) *E. coli* cells, resuspended in buffer A (50 mM Tris pH 8, 150 mM NaCl, 0.05% NP40) and lysed by adding lysozyme 10 mg/mL, 1 mM EDTA and protease inhibitor.

After sonication, the clear cell lysates corresponding to 10 µg of HALO-DDX3X or MBP-Np proteins were bound to 30 µL of HALO-link resin (Promega, MD, USA) or 30 µL of MBP resin Amintra (Abcam, Cambridge, UK), respectively, and incubated for 4 h at 4 °C at 850 rpm on a Thermomixer HC (Starlab srl, Milan, Italy). Then, after extensive washing with 1x PBS, 10 µg of purified Np or DDX1 proteins were diluted to 150 µL in buffer A, added to the resin and incubated overnight at 16 °C on Thermomixer HC. The following day, the supernatant was discarded, and the resin was washed with 1X PBS and treated with 5U of TEV protease (Promega, MD, USA) in 150 µL of buffer A plus 1 mM DTT for 2 h at 25 °C in agitation. Finally, the supernatant was recovered and the collected samples were analyzed using Western blot analysis. Control experiments were carried out by using BL21 (DE3) cell lysates which had been transformed with pH6HTN-His6-Halotag or pET HIS6-MBP-TEV empty vectors.

### 4.10. Non Denaturing Gel-Based Helicase Assays

The helicase activity of DDX1 and DDX3X was monitored by measuring the conversion of 50 nM dsRNA 18/38 mer FAM into a ssRNA 18 mer FAM. Reactions were performed in a helicase buffer (20 mM Tris–HCl pH 8, 2 mM DTT, 70 mM KCl, 2 mM MgCl_2_, 6U RNasin) and started with the addition of 4 mM ATP. After 30 min of incubation at 37 °C, reactions were stopped by adding 6x gel loading buffer (Thermo Fisher Scientific, Waltham, MA, USA) and run on 10% TBE-Polyacrylamide 19:1 + 0.1% SDS gel at 40 V for about 3 h in a 1X TBE + 0.1% SDS buffer at 4 °C in mini-PROTEAN electrophoresis system (Bio-Rad, Hercules, CA, USA). Substrates and products were quantified by laser scanning densitometry at Typhoon-TRIO (GE Healthcare, Uppsala, Sweden). When Np was used in the presence of DDX1 or DDX3, proteins were preincubated for 10 min at 4 °C before the addition of dsRNA and ATP.

## 5. Conclusions

In summary, the results presented here show for the first time, to the best of our knowledge, a direct interaction of SARS-CoV-2 with the host DDX1 and DDX3X helicases and their reciprocal effects on RNA interaction. As discussed above, the viral Np as well as the host DDX1 and DDX3X modulate several important aspects of the viral life cycle and of the host immune response. The novel functional interaction among these proteins described here not only adds to our understanding of the virus–host interface at the molecular level, but can also suggest innovative therapeutic approaches. For example, protein–protein interaction inhibitors could be designed to disrupt the binding of Np to DDX1 and/or DDX3X. Additionally, PROTACs constructs could be developed based on the interaction domains among these proteins to specifically target the Np–DDX1/–DDX3X complexes for degradation.

## Figures and Tables

**Figure 1 ijms-24-05784-f001:**
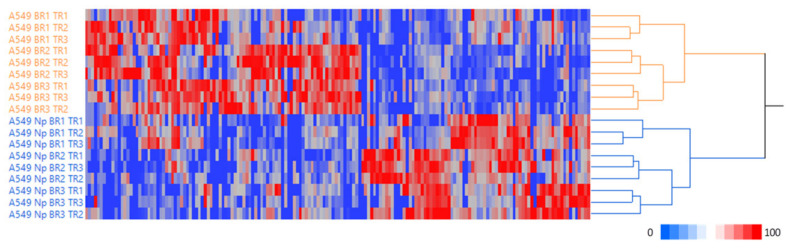
Hierarchical clustering of single protein lists from A549 cells transfected with an empty vector and with SARS-CoV-2 N protein. The dendogram was obtained by computing the peptide spectrum matches (PSMs) of statistically significant proteins selected by linear discriminant analysis (LDA); Euclidean’s distance metric and Ward’s methods were applied (JMP15.2 software). Categories were reported in different colours accordingly to the strain considered.

**Figure 2 ijms-24-05784-f002:**
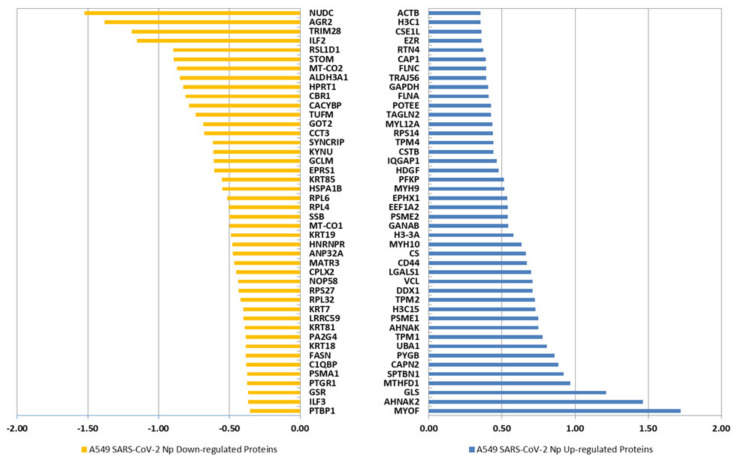
Differentially expressed proteins identified in A549 cells expressing an empty vector and SARS-CoV-2 N protein through label-free quantification with MAProMa software. Orange bars and negative DAve values refer to down-regulated proteins in A549 transfected with SARS-CoV-2 N protein, while light blue bars and positive DAve values refer to up-regulated proteins in A549 cells transfected with SARS-CoV-2 N protein. For each protein, the gene name and related DAve value (ratio of protein expression) are reported.

**Figure 3 ijms-24-05784-f003:**
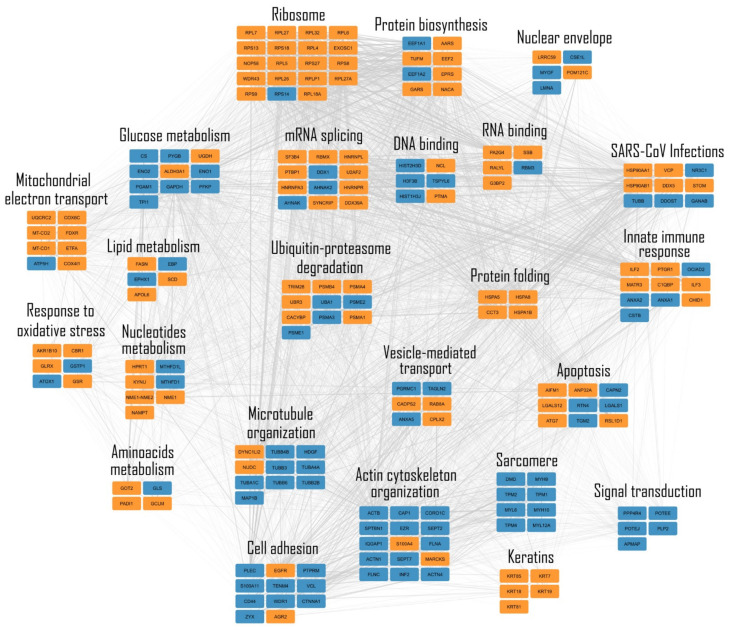
Protein–protein interaction (PPI) network of differentially expressed proteins in A549 cells with empty vectors and A549 cells transfected with SARS-CoV-2 N protein. Physical or/and functional interactions are highlighted by thicker edges and in consideration of experimental (STRING score > 0.15) and database (STRING score > 0.35) annotated interactions. The networks were visualized by Cytoscape v.3.9.1 software, while biological processes were retrieved by STRING enrichment. The color code of distinct nodes reflects the differential expression of a protein in the examined conditions (orange for up-regulated in A549 with empty vector and blue light for up-regulated in A549 with SARS-CoV-2 N protein).

**Figure 4 ijms-24-05784-f004:**
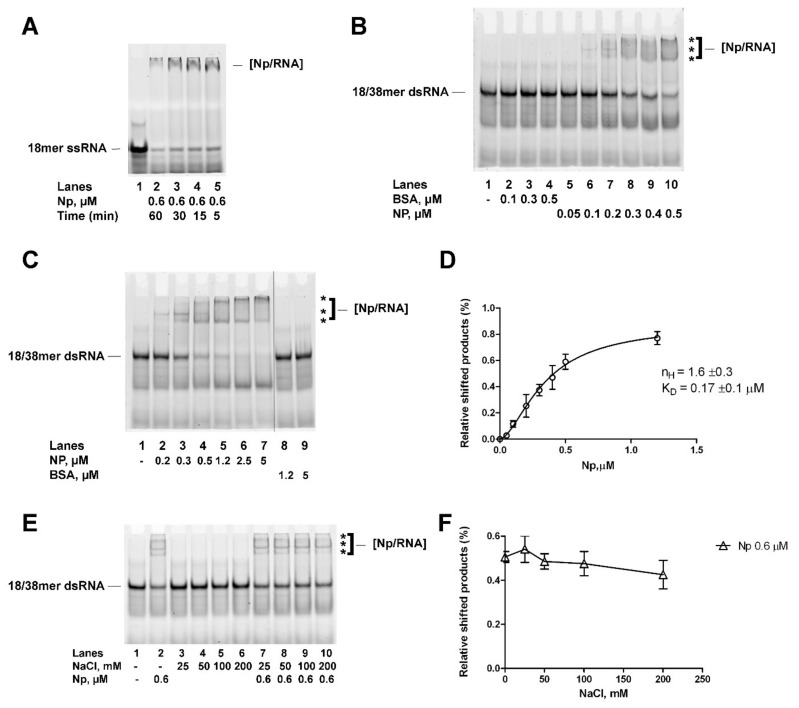
SARS-CoV-2 Np protein cooperatively binds dsRNA. Asterisks (***) in all panels indicated the high molecular weight Np-dsRNA complexes. (**A**). EMSA showing the time-course of recombinant Np binding to ssRNA. Lane 1, control in the absence of protein. (**B**). EMSA showing the titration of BSA (lanes 2–4) or recombinant Np (lanes 5–10) in the presence of dsRNA. Lane 1, control in the absence of protein. (**C**). EMSA showing the titration of recombinant Np (lanes 2–7) or BSA (lanes 8, 9) in the presence of dsRNA. Lane 1, control in the absence of protein. (**D**). Variation in dsRNA-Np complex formation as a function of Np concentration. Curves were fitted according to a cooperative model with the software GraphPad Prism 6.0. Data points represent the mean of two independent experiments ± S.E. (**E**). EMSA showing the titration of NaCl in the presence of dsRNA and in the absence (lanes 3–6) or in the presence (lanes 7–10) of recombinant Np. Lane 1, control in the absence of protein; Lane 2, positive control in the presence of Np and in the absence of NaCl. (**F**). Variation in dsRNA–Np complex formation as a function of NaCl concentration. Data points represent the mean of two independent experiments ±S.E.

**Figure 5 ijms-24-05784-f005:**
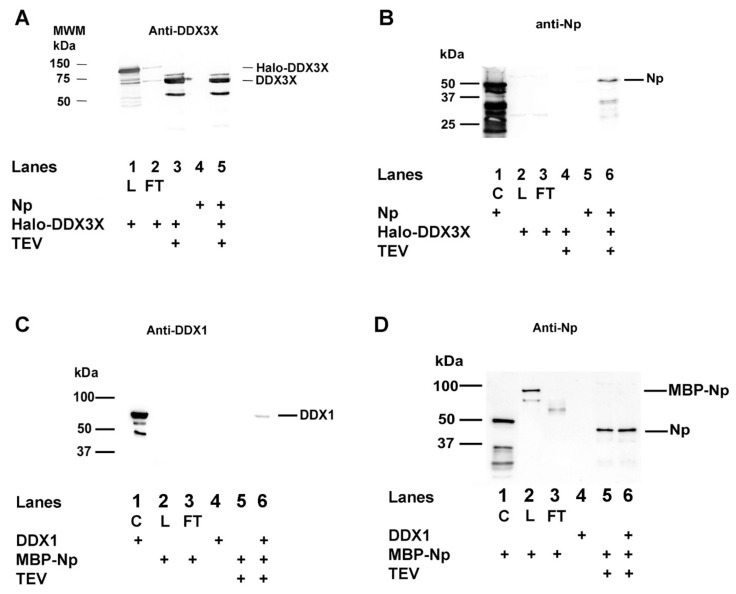
Np protein physically interacts with human DDX3X and DDX1. (**A**). Western blot with anti-DDX3X Abs of the fractions eluted from the HALO affinity beads with TEV protease. Lane 1, loading control of recombinant HALO-tagged DDX3X crude extract; Lane 2, flow through; Lane 3, DDX3X incubated with beads in the absence of Np; Lane 4: Np incubated with the Halo tag captured on the beads in the absence of DDX3X; Lane 5: DDX3X incubated with the beads in the presence of Np. (**B**). Western blot with anti-Np Abs of the same fractions eluted from the HALO affinity beads shown in panel A. Lane 1: positive control for the Abs with recombinant Np alone; Lane 2, loading control of *E. coli* expressing recombinant Halo-tagged DDX3X crude extract; Lane 3, flow through; Lane 4, DDX3X incubated with beads in the absence of Np; Lane 5: Np incubated with the Halo tag captured on the beads in the absence of DDX3X; Lane 6: DDX3X incubated with the beads in the presence of Np. (**C**). Western blot with anti-DDX1 Abs of the fractions eluted from the MBP affinity beads with TEV protease. Lane 1, control with recombinant DDX1; Lane 2, loading of *E. coli* expressing recombinant MPB-tagged Np crude extract; Lane 3, flow through; Lane 4, DDX1 incubated with beads in the absence of MBP-tagged Np; Lane 5: Np incubated with the beads in the absence of DDX1; Lane 6: DDX1 incubated with the beads in the presence of MBP-tagged Np. (**D**). Western blot with anti-Np Abs of the fractions eluted from the MBP affinity beads with TEV protease shown in panel C. Lane 1, control with purified recombinant untagged DDX1; Lane 2, loading of *E. coli* expressing recombinant MPB-tagged Np crude extract; Lane 3, flow through; Lane 4, DDX1 incubated with MBP tag captured on the beads in the absence of Np; Lane 5, MPB-tagged Np incubated with beads; Lane 5, MBP-tagged Np incubated with the beads in the presence of DDX1.

**Figure 6 ijms-24-05784-f006:**
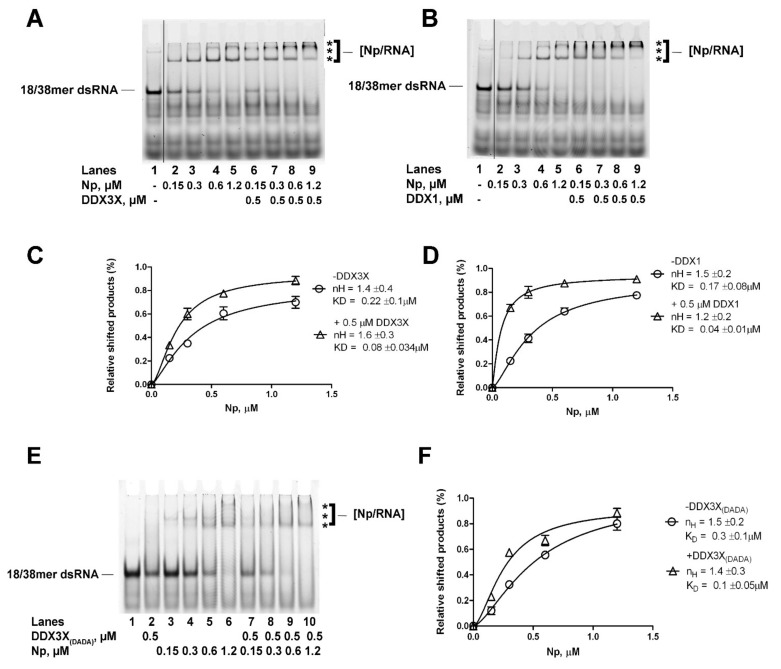
DDX1 and DDX3X increase the affinity of SARS-CoV-2 Np protein for dsRNA. Asterisks (***) in all panels indicated the high molecular weight Np–dsRNA complexes. (**A**). EMSA showing the titration of Np in the presence of dsRNA and in the absence (lanes 2–5) or in the presence (lanes 6–9) of DDX3X. Lane 1, control in the absence of proteins. (**B**). As in panel A, but with DDX1 replacing DDX3X. (**C**). Np–dsRNA complex formation as a function of Np concentrations in the absence (circles) or in the presence (triangles) of DDX3X. Curves were fitted according to a cooperative model with the software GraphPad Prism 6.0. Data points represent the mean of two independent experiments ± S.E. (**D**). As in panel C but with DDX1 replacing DDX3X. (**E**). EMSA showing the titration of Np in the presence of dsRNA and in the absence (lanes 3–6) or in the presence (lanes 7–10) of the catalytically dead mutant DDX3X_(DADA)_. Lane 1, control in the absence of proteins. Lane 2 control in the presence of DDX3X_(DADA)_ alone. (**F**). Np–dsRNA complex formation as a function of Np concentrations in the absence (circles) or in the presence (triangles) of the catalytically dead mutant DDX3X_(DADA)_. Curves were fitted with the software GraphPad Prism 6.0 according to a cooperative model. Data points represent the mean of two independent experiments ± S.E.

**Figure 7 ijms-24-05784-f007:**
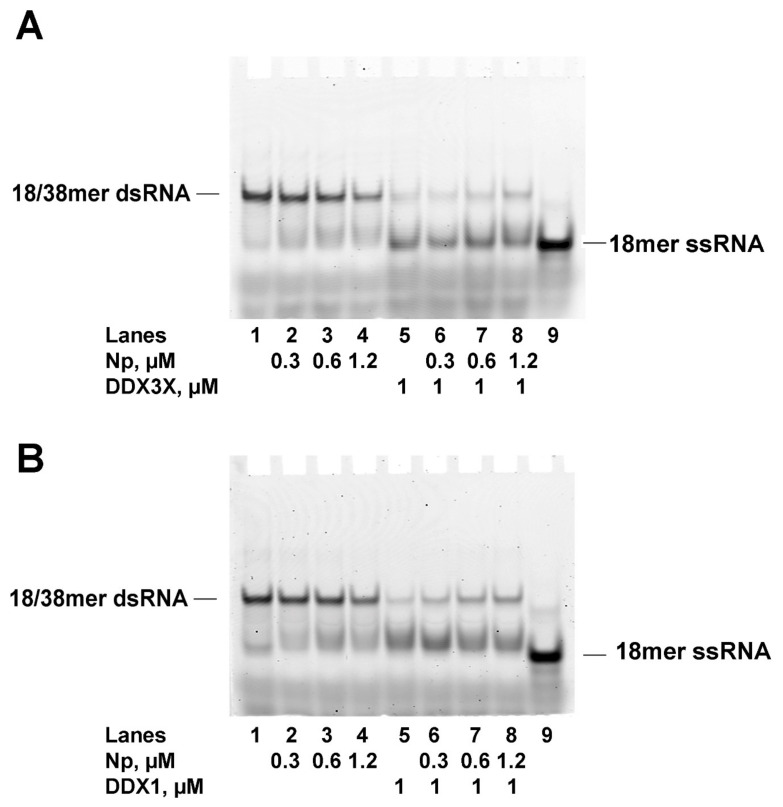
SARS-CoV-2 Np protein inhibits the RNA helicase activity of DDX3X and DDX1. (**A**). Native PAGE analysis of the helicase reaction in the presence of Np alone (lanes 2–4), DDX3X alone (lane 5) or a combination of both (lanes 6–8). Lane 1, control 18/38 mer dsRNA incubated in the absence of proteins. Lane 9, control 18 mer ssRNA incubated in the absence of proteins. (**B**). Native PAGE analysis of the helicase reaction in the presence of Np alone (lanes 2–4), DDX1 alone (lane 5) or combination of both (lanes 6–8). Lane 1, control 18/38 mer dsRNA incubated in the absence of proteins. Lane 9, control 18 mer ssRNA incubated in the absence of proteins.

## Data Availability

The proteomic datasets (in form of raw data) analyzed for this study can be found in the MassIVE database at the link ftp://massive.ucsd.edu/MSV000090979/ (accessed on 14 March 2023).

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
