# Peer review of "Interaction of SARS-CoV-2 Nucleocapsid Protein and Human RNA Helicases DDX1 and DDX3X Modulates Their Activities on Double-Stranded RNA"

_ijms, 2023, doi:10.3390/ijms24065784_

Round 1

Reviewer 1 Report

The manuscript entitled “Interaction of SARS-CoV-2 nucleocapsid protein and human RNA helicases DDX1 and DDX3X modulates their activities on double-stranded RNA” by Lodola et al. evaluated and describe the possible role of cellular RNA helicase DDX1 and DDX3X in the life cycle of the virus. Overall, the manuscript is noteworthy and requires minor revision before publication in IJMS, as follows:

Comments

1.      Abstract, please add a few experimental details (quantitative data), and a statement of novelty and significance of the present study.

2.      Lines 29-31, please provide a few sentences on the SARs-CoV-2 properties, emergence, and significant challenges due to their significant evolution as various variants in a very short period, i.e., Infection 50 (2022) 309–325.

3.       Line 102, space between value and unit. Please do this in other places in the text.

4.      Please add one illustration based on the summary of the present finding.

5.      Discussion can be a minor upgrade with some quantitative data results.

6.      Conclusions, these can be more elaborated along with advantages and future perfectives. 

Author Response

Reviewer 1

The manuscript entitled “Interaction of SARS-CoV-2 nucleocapsid protein and human RNA helicases DDX1 and DDX3X modulates their activities on double-stranded RNA” by Lodola et al. evaluated and describe the possible role of cellular RNA helicase DDX1 and DDX3X in the life cycle of the virus. Overall, the manuscript is noteworthy and requires minor revision before publication in IJMS, as follows:

Q1. Abstract, please add a few experimental details (quantitative data), and a statement of novelty and significance of the present study.

A1. We thank the Reviewer for this suggestion. We have now revised the Abstract accordingly

Q2.Please provide a few sentences on the SARs-CoV-2 properties, emergence, and significant challenges due to their significant evolution as various variants in a very short period, i.e., Infection 50 (2022) 309–325.

A2. We are grateful to the Reviewer for this important remark. We have now added at the beginning of the Introduction some sentences highlighting the points requested by the Reviewer.

Q3. Line 102, space between value and unit. Please do this in other places in the text.

A3. We apologize for the typing errors. The text has been fully revised as suggested.

Q4. Please add one illustration based on the summary of the present finding.

A4. We thank the Reviewer for the suggestion. We have now prepared a graphical abstract which summarise the main findings of the manuscript.

Q5. Discussion can be a minor upgrade with some quantitative data results.

A5. We thank the Reviewer for this suggestion. However, we were trying to avoid to repeat too extensively the quantitative data already reported in the Results section in order to give more space to the interpretation of the data in a broader context. We have nonetheless briefly mentioned the -fold stimulation of Np RNA binding affinity.

Q6. Conclusions, these can be more elaborated along with advantages and future perfectives.

A6. We agree with the Reviewer that our previous version was too succinct. We have now expanded the Conclusions, suggesting possible strategies that could emerge from the exploitation of our results.

Reviewer 2 Report

The study presented by Lodola et al. investigates the changes in the proteome after Sars-CoV2 Np overexpression in A549 cells and identifies amongst the affected proteins two DEAD-box RNA helicases DDX3X and DDX1 as promising candidates to take forward into more detailed characterisation. Using electrophoretic mobility shift assays, the authors show that Np binds ssRNA as well as dsRNA, and that RNA binding by Np is stimulated in the presence of either helicase regardless of the helicases’ catalytic capacity. At the same time, they find that the RNA unwinding activity the helicases becomes completely inhibited in the presence of Np. Thus they conclude a model in which interaction of Np with DDX1 or DDX3X results in mutual modulation of their activities, stimulation for Np, and inhibition for the helicases.

The proteomics approach is elegant and the biochemical experiments characterising RNA binding and unwinding by Np, DDX1, DDX3X are mostly supportive of the interpretations and conclusion made by the authors. However, there are a few instances of concerns about missing controls, inappropriate quantification, number of repeat experiments and inconsistencies between experiments, that need to be addressed in order to convincingly support the conclusions made.

most critical points

-          The “dsRNA” they use contains a single stranded overhang, and ssRNA binding by Np has been shown. To conclude Np truly binds dsRNA, a blunt ended dsRNA must be used

-          The general quality of the gels is low and resolution of RNA-Np complexes very low and often extremely close to the wells of the gel. In some cases, some complexes might not even be resolved/or migrated into the gel leading to misinterpretation of the gels, e.g. Fig6A lanes 2+3 and 6B lanes 2+3. Do you have any gels that were run for a longer period of time? As a general suggestion, you could use fluorescence polarisation as a second approach using the same samples even to confirm your results.

-          The IP experiment in Fig5 lacks a negative control of using pre-cleaved bait together with the prey. Further, detection of the proteins has to be shown at least once from the very same membrane and not separately as currently presented

-          Figure 6 seems to have been done only as 1 repeat, and quantification is very questionable as in my opinion the gel does not reflect the quantification shown. In addition, there are high molecular weight complexes in the wells of 6B lanes 2+3. Does that perhaps contribute to the interpretation of lanes 7-10? what does it mean?

-          Figure 6 and 7 are instances of similar experiment but quality appears extremely different

-          Figure 7 lacks control of DDX-only lane, which is required for quantification in Fig 7C+D. Thus curves for Np binding in presence of DDXs is currently normalised inappropriately

-          A control is missing that DDX3X-DADA is inactive

-          Figure 7 panels A-B and F show strong inconsistencies of  Np binding in the absence of helicase. A-B show binding up to 80-100%, F only to 20%. This has been done in duplicate. How do you explain this discrepancy.

-          In Figure 8 is the RNA signal not consistent across the lanes which, in the absence of loading controls, can lead to misinterpretation of the results. It seems this EMSA is run at the same conditions as the binding EMSA except addition of ATP. Wouldn’t you perhaps expect to see bands corresponding to Np or DDX RNA binding in the gel? This could be a reason for the apparent inhibition of helicase activity while it is actually that the proteins are bound to the products of the reaction. Unless the RNA signals are consistent between the lanes, or a loading control is shown, this cannot be interpreted. Good control would be to repeat in the presence of the DADA mutant and to run the gel under protein-denaturing conditions to show consistency of the RNA signal

-          There is no references given for the N-interactor dataset they used to compare line 335, nor to “previous result” line 361

minor

-          include all column names in Stable 2 and explanantions

-          proteomics methods: What are each samples normalised to? IT could be good to explain briefly why these DAve and DCI cut offs were chosen. How long was Np overexpressed for?

-          In Figure 2 the presentation could be misleading using the terms “Empty Up-regulation”. Easier would be to use “downregulated in NP expression” as commented in the text

-          Figure 3 is very busy and really hard to see. I suggest to just use an GO term analysis which focuses on what the authors are already only focussing on in the text.

-          Figure 4: How is the experiment performed at the different salt concentrations? What is the storage buffer of Np? It might likely be that given the duration of run and conditions that an EMSA is not the best approach to test this. What about fluorescence spectroscopy? intensity or polarisation should work

Author Response

Reviewer 2

The proteomics approach is elegant and the biochemical experiments characterising RNA binding and unwinding by Np, DDX1, DDX3X are mostly supportive of the interpretations and conclusion made by the authors. However, there are a few instances of concerns about missing controls, inappropriate quantification, number of repeat experiments and inconsistencies between experiments, that need to be addressed in order to convincingly support the conclusions made.

Most critical points

Q1. The “dsRNA” they use contains a single stranded overhang, and ssRNA binding by Np has been shown. To conclude Np truly binds dsRNA, a blunt ended dsRNA must be used.

A1. We thank the Reviewer for this observation. Indeed, the substrate we used has a short 5’-ss oh. The reason was that this is the optimal substrate for testing the RNA helicase activity of DDX1 and DX3X, while their unwinding activity on blunt dsRNA is low. We would like to point out that our main aim was not to demonstrate that Np binds dsRNA. This has been already shown by others in different publications (se Refs. 1-6 of th manuscript). We, on the other hand, were interested in testing the effects of DDX1 and DDX3X on Np and vice versa. For this reason, we had to use a substrate which could be utilized by all three proteins. Our experiments show that Np can productively bind this partially dsRNA. We have now more clearly explained this point in the revised manuscript (lines 372-374). In addition, we have now consistently indicated the substrate used throughout the manuscript as “partially dsRNA”, in order to resolve this ambiguity.

Q2-4. The general quality of the gels is low and resolution of RNA-Np complexes very low and often extremely close to the wells of the gel. In some cases, some complexes might not even be resolved/or migrated into the gel leading to misinterpretation of the gels, e.g. Fig6A lanes 2+3 and 6B lanes 2+3. Do you have any gels that were run for a longer period of time? As a general suggestion, you could use fluorescence polarisation as a second approach using the same samples even to confirm your results. Quantification of free RNA

Figure 6 seems to have been done only as 1 repeat, and quantification is very questionable as in my opinion the gel does not reflect the quantification shown. In addition, there are high molecular weight complexes in the wells of 6B lanes 2+3. Does that perhaps contribute to the interpretation of lanes 7-10? what does it mean? Figure 6 and 7 are instances of similar experiment but quality appears extremely different

A2-4. We thank the Reviewer for this observation. Unfortunately, we do not have access to fluorescence polarization. However, in our analysis we did not look for precise quantification of the different high molecular weight forms, but quantified only the relative intensities of the overall shifted products with respect to the unbound substrate. Quantification was done expressing the activity as relative shifted products, according to the formula S/(U+S) were U and S are the intensities of the Unbound and Shifted RNA molecules, respectively. This also allowed as to account for eventual gel loading fluctuations, since the shifted products in each lane were internally normalized with respect to the total.

Any signal in the well, that is which did non enter the gel, was not considered in the quantification.

As correctly pointed out by the Reviewer, the quantifications shown in the original Figure 6 were done for a single experiment. Th experiments shown in the original Figure 6 were preliminary attempts to test for any possible effect of the combination of Np with DDX1 or DDX3X. We agree that they are at best semi-quantitative. The experiments shown in original Figure 7 with a more careful kinetic analysis were indeed the quantitative ones. So, in the revised manuscript we have moved the original Figure 6 as the Suppl. Figure S3, indicating it in the main text as a set of preliminary experiments (lines 448-450) and avoiding to draw any further conclusion.

Q5-6. Figure 7 lacks control of DDX-only lane, which is required for quantification in Fig 7C+D. Thus, curves for Np binding in presence of DDXs is currently normalised inappropriately (se Fig 6 with no quantification)

Figure 7 panels A-B and F show strong inconsistencies of Np binding in the absence of helicase. A-B show binding up to 80-100%, F only to 20%. This has been done in duplicate. How do you explain this discrepancy?

A5-6. We thank the Reviewer for this observation. As far as the control with DDX1 and DDX3 alone, in all the experiments done for the manuscript neither DDX1 nor DDX3X were able to shift the RNA substrate. For example, see the Supp. Figure S3 of the revised manuscript. However, in response to the Reviewer’s request, we have now included as Supp. Figure S4 two experiments showing the very same titration as shown in the original Figures 7 A, B (now Figures 6 A, B of the revised manuscript), but including the points with DDX1 or DDX3X alone.

We absolutely agree with the Reviewer that the activity of Np as reported in the original Figure 7 E was lower than the one shown in the panels A and B. This was due to the fact that the experiments have been performed in different times and using a fresh preparation (panels A and B) in one case and the same but stored for a few weeks in freezer in the other case (panel E). In order to be more consistent, we have now repeated the experiment (two replicates) with the DDX3X DADA mutant with a fresh preparation of Np (new Figure 6 E of the revised manuscript), now showing comparable activity throughout all the experiments (see the new Figures 6 A, B and E). Figure 6 F now shows the new quantification made out of the two replicates, which shows consistent results with respect to previous one, with only marginal differences (as expected) in the absolute values of the KD and nH parameters.

Q7. The IP experiment in Fig5 lacks a negative control of using pre-cleaved bait together with the prey. Further, detection of the proteins has to be shown at least once from the very same membrane and not separately as currently presented

A7. We thank the Reviewer for bringing up this point, which allowed us to realize an apparent lack of clarity of the manuscript. Indeed, as originally indicated in Materials and Methods the negative controls of the pulldown experiments were performed by incubating the prey proteins together with lysates that contain the control plasmids expressing the Halo tag-tev or the MPB-tev tags alone as bait proteins. These reactions were conducted with the same procedures of our IP samples and actually represent the best negative control to verify the pulldown assay.

In the results of the revised manuscript, we have modified the text and the legend of Figure 5 to highlight this point.

Moreover, the IP were carried out by dividing each sample in equal parts at the end of the experiment to exactly load the different gels with the same reaction mixture. Pulldown reactions were performed more than once always obtaining the same interaction described in the text.

Q8. A control is missing that DDX3X-DADA is inactive

A8. We thank the Reviewer for this remark. The DADA mutant has been generated and characterized by our group several years ago. We have added a reference to a publication of ours where it is shown that the mutant is catalytically inactive (Ref. 39 of the revised version).

Q9. In Figure 8 is the RNA signal not consistent across the lanes which, in the absence of loading controls, can lead to misinterpretation of the results. It seems this EMSA is run at the same conditions as the binding EMSA except addition of ATP. Wouldn’t you perhaps expect to see bands corresponding to Np or DDX RNA binding in the gel? This could be a reason for the apparent inhibition of helicase activity while it is actually that the proteins are bound to the products of the reaction. Unless the RNA signals are consistent between the lanes, or a loading control is shown, this cannot be interpreted. Good control would be to repeat in the presence of the DADA mutant and to run the gel under protein-denaturing conditions to show consistency of the RNA signal

A9. We thank the Reviewer for these remarks. We would like firstly to precise that the conditions for the non denaturing gels used in the helicase assays were different from those employed for the EMSA. Specifically, the helicase reactions were performed at 37°C while EMSA reactions were incubated at 4°C and the helicase gels contained 0.1% SDS. We have added the precise conditions in the M&M section of the revised manuscript. However, we agree that the gels shown in the original Figure 8 were not optimal. So, we performed new helicase assays, now shown in the new Figure 7 of the revised manuscript, titrating Np alone or in the presence of DDX1 or DDX3X. Now the signals are very consistent across the lanes and show a Np-dependent inhibition of the helicase activity with no supershifted products, thus ruling out Np-RNA complex formation under the running conditions in the presence of SDS.

Q10. There are no references given for the N-interactor dataset they used to compare line 335, nor to “previous result” line 361

A10. We apologize for this oversight. We have now added the relevant references.

Minor points

  1. include all column names in Stable 2 and explanations

A: We apologize for an error occurred during the first manuscript submission due to which had not been loaded the final version of some supplementary tables. Now the issue has been fixed and all supplementary tables with their column headers and legends are correctly uploaded as supporting information.

Q: proteomics methods: 1) What are each samples normalised to? 2) It could be good to explain briefly why these DAve and DCI cut offs were chosen. 3)How long was Np overexpressed for?

A: 1) As reported in M&M section, line 152 ref.25, and in Results section, line 288 ref 33, all protein lists were aligned and normalized using a total signal normalization method (Carvalho PC, Fischer JS, Chen EI, Yates JR, Barbosa VC. PatternLab for proteomics: a tool for differential shotgun proteomics. BMC Bioinformatics 2008;9:316.). According to this method, given two or more sample S1, S2...Si, and T1, T2...Tj the respective sum of PSMs for all proteins identified, the goal is to obtain T1 = T2…… = Tj. Therefore, normalization of PSMsij was obtained as follows: nPSMsij = PSMsij/Tj, where Tj is the value obtained by summing PSMs values of all proteins belonging to the same list. The information requested by the Referee is now provided in M&M section of  the revised manuscript: “Then, all protein lists were aligned, normalized with total signal normalization method and compared by means of the average peptide spectrum matches (aPSM) [25,26], corresponding to the average of all the spectra identified for a protein and, consequently, to its relative abundance, in each analyzed condition.”

2) As reported in M&M section, DAve and DCI are two algorithms included in the home-made MAProMa software for label free quantitation (De Palma et al., mBio 2020; Riccio et al., WAO Journal 2020; Sereni et al., JACI 2018; Mauri et al. Methods Enzymol. 2008) that are applied on the aPSMs of proteins belonging to the two examined groups, already statistically filtered and averaged, in order to extract differentially expressed proteins. Based on a direct correlation between PSMs and the relative abundance of the identified proteins, DAve evaluates changes in protein expression and it is defined as (X-Y)/(X+Y)/0.5, while DCI evaluates the confidence of differential expression and it is defined as (X+Y)*(X-Y)/2, where X and Y represent the aPSMs of a given protein in two compared samples. So, the thresholds of DAve ≥│0.35│ and DCI≥│5│applied, as stated at lines 295-299 of Results section, allow to consider proteins with a variation greater than a fold change of 1.5, maximizing the confidence of identification. In fact, DAve allows a value comparable to ln[FC]; of note FC requires an arbitrary value for protein exclusively identified in one sample, to avoid non-sense values (such as n/0 or 0/n); on the contrary, DAve returns always a finite value.

Unfortunately, for typing errors in the uploaded pdf version of the manuscript the thresholds of DAve and DCI were not correctly reported, causing probably a misleading interpretation. Now we fixed these issues, we added a further reference n.27 in M&M section and implemented the corresponding sentences with more details on the two indexes:

“In depth, to select differentially expressed proteins, A549 cells transfected with an empty vector and with SARS-CoV-2 N protein were compared, applying a threshold of │0.35│ and │5│ on the two MAProMa indexes DAve (Differential Average) and DCI (Differential Confidence Index), respectively. DAve, which evaluates changes in protein expression, was defined as (X – Y)/(X + Y)/0.5 with values ranging from a maximum of +2.00 to a minimum of -2.00, while DCI, that evaluates the confidence of differential expression, was defined as (X +Y) x (X – Y)/2 with values ranging from +∞ to –∞. The X and Y terms represent the aPSMs of a given protein in two compared samples.

3) Np was overexpressed for 72h. We have now added this information in th M6M section of the revised manuscript and added a Western Blot to show the level of expression as Suppl. Figure S1 A.

  1. In Figure 2 the presentation could be misleading using the terms “Empty Up-regulation”. Easier would be to use “downregulated in NP expression” as commented in the text.

  1. A. We thank the reviewer for this suggestion and replaced the histogram title with “downregulated in NP expression”.

  1. Figure 3 is very busy and really hard to see. I suggest to just use an GO term analysis which focuses on what the authors are already only focussing on in the text.

  1. We thank the Reviewer for this remark and we fully understand the Reviewer’s concern, because interpreting a PPI network could be a particularly challenging task due to its complexity, but it is important for understanding how proteins are interconnected and work together in a coordinated fashion in a cell to perform cellular functions and for highlighting pathways in which differentially expressed proteins are involved. However, to meet the Reviewer request, we added as Supp. Figure S2 the Voronoi diagrams that show, based on the KEGG Gene classification, the most enriched protein functions with polygon areas reflecting protein abundances in the two examined conditions.

  1. Figure 4: How is the experiment performed at the different salt concentrations? What is the storage buffer of Np? It might likely be that given the duration of run and conditions that an EMSA is not the best approach to test this. What about fluorescence spectroscopy? Intensity or polarisation should work

A. We thank the reviewer for this remark. NaCl was added to reaction mixture at the indicated final concentrations and reactions were incubated as described in Material and Methods. Unfortunately, we do not have access to spectroscopy assays. We agree that EMSA might not be the optimal way to assess the effects of salt. However, if the resulting ionic strength was affecting significantly the ability of Np to bind RNA or the stability of the resulting complexes, this should have been detected in terms of amounts of shifted products. The storage buffer of concentrated stock preparations of Np contained 200 mM KCl, but upon dilution in the reaction mixture by 20-fold this resulted in a final concentration of 10 mM KCl in the assay, thus not affecting the final NaCl concentration. Any possible residual effect of the salt in the storage buffer was in any case normalized, since it was equal in all the samples.

Round 2

Reviewer 2 Report

The authors addressed all points risen appropriately.